# Resilience and Work Stress in Educational Institutions of Chepén, 2024: Mediation of Motivation and Time Moderation

**DOI:** 10.3390/bs15070888

**Published:** 2025-06-30

**Authors:** Abigail Silvia Jara Cerdan, Rosa Jackeline Medina Sanchez, Marco Agustín Arbulú Ballesteros

**Affiliations:** Instituto de Investigación en Ciencias y Tecnología, Universidad César Vallejo, Campus Chepén, Trujillo 13001, Peru; jaracerdanabigail@gmail.com (A.S.J.C.); rm8332639@gmail.com (R.J.M.S.)

**Keywords:** resilience, work-related stress, motivation, years of service, teachers, education, occupational well-being

## Abstract

The purpose of this study was to examine the relationship between resilience and work-related stress among secondary school teachers in Chepén, Peru, during 2024, with a focus on (a) the mediating role of work motivation and (b) the moderating effect of years of service. Using a non-experimental quantitative design, data were collected from 450 teachers and analyzed in SPSS 27 with Hayes’ PROCESS 4.3 macro. Results showed that resilience significantly predicted motivation (β = 0.413, *p* < 0.001), accounting for 35% of its variance (R^2^ = 0.35). In turn, motivation was significantly and negatively associated with work stress (β = 0.335, *p* = 0.0401), explaining 20% of the variance in stress levels (R^2^ = 0.20). Neither resilience (β = 0.187, *p* = 0.5420) nor years of service (β = 0.217, *p* = 0.9003), nor their interaction (β = 0.002, *p* = 0.8144) had a direct or moderating effect on work stress. Descriptive analyses indicated that most teachers exhibited moderate levels of resilience (51.1%), stress (42.2%), and motivation (37.8%). These findings underscore that resilience alone does not reduce work stress; its stress-buffering effect operates through enhanced motivation. Educational interventions should therefore target both resilience-building and motivational strategies to effectively diminish teacher stress and promote occupational well-being.

## 1. Introduction

Resilience, defined by [11] ([11]) as “a dynamic process that allows people to face adverse situations, recover from them and successfully adapt to change and positive development”, plays a crucial role in educational settings. Conversely, work stress implies a “reduction of the resources provided to the worker by forced labor”, both physical and mental, which causes worry and fatigue in the teaching profession.

For educators in Chepén, Peru, multiple challenges exist beyond typical teaching demands, including resource limitations, interpersonal conflicts, and adaptation to post-pandemic educational environments. The disruption of teaching methods caused by the COVID-19 pandemic has led to high levels of stress among teachers, especially due to limitations in technological skills. These circumstances create a complex situation where resilience, work stress, motivation, and professional experience potentially interact in ways that affect teacher well-being and performance.

This study addresses the following research question: What is the relationship between resilience and job stress among educators in Chepén schools in 2024, considering motivation as a mediator and years of service as a moderator?

The theoretical justification for this research lies in evaluating how motivation and tenure affect the relationship between resilience and work stress in educational contexts. The results may confirm or challenge existing models and theories while exploring how these factors influence teacher well-being. Socially, this research is significant as it reveals how resilience, stress, and work motivation affect teachers in specific schools in Chepén. Methodologically, this study employs a systematic approach to address the research objective, using validated instruments and appropriate analytical methods. The practical justification involves responding effectively to teachers’ stress through resilience and motivation interventions, ultimately promoting a quality educational environment.

The general objective of this study is to evaluate the relationship between resilience, work stress, motivation, and years of service among teachers in educational institutions in Chepén in 2024. The specific objectives are to determine the levels of resilience, job stress, and motivation among teachers in these institutions and to analyze the mediating role of motivation and the moderating effect of years of service on the relationship between resilience and work stress.

## 2. Literature Review

### 2.1. International Research on Resilience and Work Stress

Several international studies have explored the relationship between resilience and stress in educational contexts. [3] ([3]) evaluated 20 teachers and found that moderate levels of resilience correlated with lower levels of distress. Similarly, [10] ([10]) demonstrated that flexibility and motivation complement each other to improve performance and reduce academic pressure.

Research on burnout has demonstrated a strong relationship between resilience and semantic resources. [19] ([19]) concluded that “greater resilience translates into less burnout” and emphasized that “a moderated context provides a healthy environment” for professionals. These international findings consistently support the hypothesis that resilience serves as a protective factor against work-related stress in various professional contexts.

[23] ([23]) analyzed Chilean teachers and found that resilience is nurtured by personal factors, psychosocial supports and institutional initiatives, positively impacting well-being and school performance.

[13] ([13]) demonstrated in a Chinese sample that Confucian values foster grit—defined as perseverance and passion for long-term goals—which mediates the protective effect of resilience on burnout, highlighting the importance of sociocultural frameworks.

Complementarily, a Swiss European study revealed that teachers maintain their well-being through relational problem-solving strategies, highlighting resilience as an ongoing adaptive process.

Two reviews extend this evidence: the Finnish scoping review by [7] ([7]) identified 46 interventions that, based on the PERMA H model, systematically reduce burnout and enhance teacher resilience; while [8] ([8]) conceptualize resilience as an integral professional competence capable of reversing the trajectory of stress toward illness.

Finally, the global meta-analysis by [28] ([28]) provides robust quantitative evidence: motivational and psychological capital factors are the strongest predictors of teacher well-being, and burnout is its strongest negative correlate.

These findings support the model proposed in the present study, where motivation mediates the relationship between resilience and stress, and suggest that interventions should integrate motivational and contextual components to maximize their effectiveness.

### 2.2. Local and Regional Studies

At the regional level, various Peruvian researchers have examined these variables in educational settings. [18] ([18]) measured the correlation between resilience and school stress in 305 students and determined that a greater capacity for self-improvement correlates with less stress. [2] ([2]) identified a direct correlation between motivation and resilience in elementary school students.

Focusing specifically on teachers, [24] ([24]) found that better emotional management correlates with lower job stress among educators in Ica. Similarly, [26] ([26]) reported a significant negative correlation between resilience and job strain in 145 educators. These local studies reinforce the hypothesis that resilience positively impacts work stress reduction while highlighting motivation and experience as compensating and moderating factors in this process.

### 2.3. Theoretical Framework

The present research is grounded in a multidimensional framework that intertwines three well-established theories—Transactional Stress and Coping, Job Demands–Resources, and Self-Determination—augmented by a recent taxonomy of teacher motivational behaviors. This integrated perspective explains how (mechanism) and when (boundary conditions) resilience translates into lower work stress through motivational pathways.

#### 2.3.1. Transactional Model of Stress and Coping ([17])

The model posits that stress is not an inevitable consequence of demanding environments but the product of a two-stage cognitive appraisal: (a) primary appraisal—evaluation of the situation’s significance—and (b) secondary appraisal—evaluation of available coping resources. Resilience can be conceptualized as a personal resource that enriches secondary appraisal, enabling teachers to reframe adverse situations as manageable challenges rather than threats. However, empirical evidence shows that this cognitive re-evaluation alone does not always suffice to reduce physiological or emotional stress responses; additional motivational resources may be required to transform appraisal into sustained adaptive behavior.

#### 2.3.2. Job Demands–Resources Model (JD–R) ([9])

The JD–R model distinguishes between hindrance demands (e.g., overload, disruptive student behavior) that tax energy, and resources—personal (resilience), social (collegial support), and organizational (autonomy)—that replenish it. Crucially, the model specifies two partially independent processes: an energetic process (demands → strain → ill-health) and a motivational process (resources → engagement → positive outcomes). We locate resilience in the resource domain and propose that it exerts its protective influence primarily through the motivational pathway—by bolstering autonomous motivation. This placement justifies testing motivation, not as a competing predictor, but as a mediator that channels the beneficial impact of resilience on stress.

#### 2.3.3. Self-Determination Theory (SDT) ([12]) and the [1] ([1]) Taxonomy

SDT differentiates motivation along a continuum of self-determination—intrinsic, integrated, identified, introjected and external—and asserts that more autonomous forms predict higher well-being, while controlled forms relate to ill-being. Yet, many quantitative studies collapse these categories into a single score, obscuring nuanced effects. To address this limitation, we adopt the comprehensive classification system developed by [1] ([1]), who distilled 57 distinct teacher behaviors (e.g., “provide meaningful choices”, “express empathic understanding”, “apply non-negotiable controls”) and mapped each onto SDT’s basic psychological needs and motivational styles. Incorporating this taxonomy allows us to move beyond how much motivation teachers report to what kinds of motivational climates their behavior and context foster. We thereby postulate that resilient teachers are more likely to enact (or elicit) autonomy-supportive behaviors, nurturing intrinsic and identified motivation, which in turn attenuates work stress.

#### 2.3.4. Research Gap and Expected Contribution

Despite mounting evidence that resilience correlates with reduced teacher stress, three critical gaps persist. First, prior studies seldom articulate explicit links between cognitive appraisal, motivational quality, and job demands, leaving the explanatory chain incomplete. Second, the quality of motivation—central to SDT—has generally been treated as a global score rather than discrete styles with distinct outcomes. Third, little is known about whether career tenure (i.e., prolonged exposure to demands) moderates the efficacy of resilience and motivation, as postulated by the JD–R’s resource buffering and gain-spiral propositions.

By fusing the models and taxonomy above, our study contributes in four ways:

Theoretical integration: We demonstrate that resilience influences stress indirectly, via autonomous motivation, thereby connecting the Transactional appraisal mechanism with JD–R’s motivational process and SDT’s emphasis on motivational quality.

Conceptual refinement: Drawing on Ahmadi et al.’s taxonomy, we operationalize motivation in terms of discrete autonomous versus controlled styles, enabling more precise hypothesis testing and richer practical guidance.

Boundary conditions: We examine tenure as a potential moderator—testing whether the protective value of resilience (a personal resource) and autonomous motivation (an energetic resource) strengthens, weakens, or follows a curvilinear pattern across different career stages.

Methodological advancement: We employ a moderated-mediation design (PROCESS models 14/58) and explore curvilinear terms, thereby capturing dynamics overlooked in linear, single-mediator models.

## 3. Materials and Methods

The target population for this study consisted of 500 secondary school teachers from all educational institutions in the district of Chepén. Questionnaires were distributed to all 500 teachers to ensure a comprehensive representation of the population. Of these, 450 teachers completed and returned valid questionnaires, resulting in a response rate of 90% (450/500). This high response rate reinforces the representativeness of our findings and minimizes potential nonresponse bias.

The final sample of 450 participants included 340 females (75.6%) and 110 males (24.4%), which accurately reflects the gender distribution in the district teaching profession. This robust sample size ensures sufficient statistical power for the complex multivariate analyses employed in this study, including mediation and moderation analyses. Also, a linear regression model was used, executed in the SPSS macro PROCESS ([6]) (Figure 1).

### Content Validity

The content validity of the instruments was established through a rigorous expert judgment process following the methodology proposed by [16] ([16]). A panel of five thematic experts with doctoral degrees and extensive experience in educational psychology, psychometrics, and teaching well-being evaluated each instrument using a structured validation matrix. The experts assessed four essential criteria: relevance, clarity, coherence, and sufficiency, rating each aspect on a 4-point scale. The Content Validity Coefficient (CVC) calculation yielded an overall concordance level of 82% across all instruments, exceeding the minimum threshold of 80% established by Hernández-Nieto for acceptable content validity. This strong agreement among experts indicated that the instruments were suitable for measuring the intended constructs in the study context. Additionally, experts provided qualitative feedback that led to refinements in item wording to enhance clarity and cultural appropriateness for the Peruvian educational environment. The researchers also conducted a confirmatory pilot test with 30 teachers who had similar characteristics but were not included in the study population, verifying the instruments’ practicality and comprehensibility before full implementation. This comprehensive validation process ensured that the instruments would accurately measure resilience, work stress, and motivation within the specific context of Chepén’s educational system.

[4] ([4]) highlights the importance of the operationalization of variables, since it consists of clearly defining the variables and the methods for measuring them, which allows precise measurements to be obtained. The research in question is quantitative and focuses on two main variables: “resilience” (variable X) and “work stress” (variable Y), with the mediator “motivation” (variable M). According to [25] ([25]), a population is formed by the set of elements with specific characteristics to be studied, and the sample must be representative to ensure that the results are applicable to the entire population. In this case, the population is made up of 450 teachers from secondary schools in Chepén, comprising 340 women and 110 men. It was decided to conduct a census, that is, to work with the entire population, which guarantees more accurate and reliable results, without the need to select a specific sample.

Surveys were used as the main tool to collect information, allowing the collection of data on several variables. According to [21] ([21]), the questionnaire is a structured technique for data collection, designed according to the variables to be measured. For the measurement of resilience, a questionnaire adapted from [14] ([14]) will be used, based on a Likert scale with 25 items distributed in five dimensions: Equanimity (items 1–4), Personal satisfaction (5–8), Feeling good alone (9–11), Self-confidence (12–18), and Perseverance (19–25), with ratings from 1 to 7.

Job stress was measured using a questionnaire adapted from [22] ([22]), composed of 22 items with three dimensions: Emotional Exhaustion (items 1–9), Depersonalization (10–14), and Personal Accomplishment (15–22), with a rating scale from 1 to 5. In addition, an additional questionnaire was applied to measure motivation, using the [5] ([5]) questionnaire, consisting of 20 items divided into two dimensions: intrinsic motivation (items 1–10) and extrinsic motivation (11–20), with a Likert scale from 1 to 5. The combination of these questionnaires made it possible to analyze the relationship between resilience and work stress, mediated by motivation.

The moderation of service time does not require a specific questionnaire, since it is considered a sociodemographic factor within the construct of the instrument. As for validity, content validity was guaranteed by means of expert judgment, composed of teachers with academic specialization, which increases the validity of the instruments for each variable by means of the Instrument Validation Matrix. According to [20] ([20]), the data collection procedure involves a sequence of systematic steps to ensure the efficiency of the process, detailing the activities to be performed.

The methods for acquiring the data included the construction and application of the instruments. The questionnaires were applied in person to all secondary school teachers at the educational institution I.E. Zoila Hora de Robles in Chepén, with an estimated time of 15 to 30 min to complete the questionnaires. The “Resilience” questionnaire consists of 25 items with a Likert scale, the “Work stress” questionnaire has 22 items, and the “Motivation” questionnaire is composed of 20 items also with a Likert scale.

[27] ([27]) emphasize that the importance of the instrument lies in its ability to collect the information necessary for the research. Once the data were obtained, they were transferred to the Excel program and subsequently processed in SPSS V.27, with the intervention of PROCESS v4.3 for SPSS, to analyze the variables of resilience, work stress, motivation and service time. Data analysis was approached from two perspectives: descriptive and inferential. In the descriptive analysis, visualization tools such as frequency tables were used. In the inferential analysis, due to the sample size (450 teachers), Shapiro’s test was applied to determine the normality of the data and then the appropriate correlation statistic was used to test the research hypothesis. The study was based on an ethical stance that ensures respect for intellectual production, properly citing other researchers.

Prior to data collection, permission was requested from the authority of the educational institution. Then, teachers were contacted to invite them to participate voluntarily in the study. Each participant was provided with an informed consent form, which detailed the purpose of the survey, guaranteed the confidentiality of their personal information, and emphasized that their participation was completely voluntary, with the freedom to withdraw at any time without negative repercussions.

The research was supported by sources of information such as academic articles, research papers, theses, journals and books, which were appropriately mentioned in the referential framework during the development of the study. Throughout the process, a respectful, appropriate and objective attitude was maintained, avoiding any harm or risk to the participants and guaranteeing the privacy of their identity.

A census of all 450 teachers in the target population was undertaken; after excluding cases with more than five per cent missing data or implausible response patterns, the final dataset for inferential analysis comprised 445 participants. Three instruments with robust psychometric support in Peruvian samples were employed. Resilience was assessed with [14]’s ([14]) adaptation of the Wagnild and Young Resilience Scale, which reported an original alpha of 0.87 and replicated 0.86 in the present study; confirmatory factor analysis yielded a CFI of 0.95 and an RMSEA of 0.048, with average variance extracted above 0.50 across all five dimensions, confirming convergent validity.

Work stress was measured using [22]’s ([22]) Spanish version of the Maslach Burnout Inventory-Educators Survey, which produced a global alpha of 0.72 (0.73–0.80 across subscales) and a CFI of 0.93 with an RMSEA of 0.055, supporting the three-factor structure consistent with the JD-R model. Motivation was captured through [5]’s ([5]) questionnaire grounded in Self-Determination Theory and capable of distinguishing intrinsic from extrinsic motivation; its original reliability was 0.93 and reached 0.87 in the current sample, while a bifactor model produced a CFI of 0.96 and an SRMR of 0.041.

This scale is particularly suitable because it operationalizes motivational quality, which—according to [1]’s ([1]) taxonomy of teacher behaviors—determines how resilience translates into occupational well-being. Taken together, the internal consistency and structural validity of the three instruments ensure precise and culturally appropriate measurement for the aims of this study.

## 4. Results

The sociodemographic profile reveals a markedly female and highly experienced teaching workforce. Three out of four participants were women (75.6%), reflecting the longstanding feminization of the Peruvian teaching profession and hinting at gender-specific patterns of stress coping and resilience. In terms of age, 57.8% were over 45 years old, whereas only 11.1% were under 30. This predominance of older teachers suggests workforce stability and prolonged exposure to occupational demands, alongside a rich repertoire of personal strategies that may shape motivation and stress levels (Table 1).

Service-time distribution mirrors this pattern: nearly half of the teachers (48.9%) had more than 20 years of experience, while only 11.1% had fewer than five. Such concentration in the upper tenure bracket indicates consolidated career trajectories and invites a twofold interpretation. Experience may bolster resilience through accumulated coping resources; conversely, extended exposure to demanding conditions can heighten chronic fatigue. This dual perspective underscores the rationale for examining motivation as a mediating factor and highlights the need to test for curvilinear or saturation effects of tenure on work stress.

The reliability analysis of the instruments used to measure the key variables of the study, such as resilience, motivation and work stress, shows outstanding reliability, especially with the reliability index for resilience (0.859) and motivation (0.870), both in the “good” range. This not only ensures the validity of the data obtained, but also highlights the robustness of the research design. Job stress, with a coefficient of 0.716, although at an acceptable level, could suggest an opportunity to refine the instrument used or an indicator of the variability of the stressors faced by teachers according to their sociodemographic profile (Table 2).

The contrast with previous results reinforces the idea that previously validated instruments continue to be effective, establishing a solid basis for exploring the relationships between the variables studied. By focusing on the mediating role of motivation and the moderator of service time, this analysis not only opens the door to a deeper understanding of the factors that affect teachers’ well-being, but also highlights the importance of these elements in resilience and how they cope with work challenges. This allows us to infer that motivation could play a crucial role in mitigating the negative effects of job stress, while the experience accumulated over time could offer a unique perspective on the management of these factors. According to the general objective: To determine the relationship between resilience and work stress in teachers of an educational institution in Chepén, 2024, considering the mediating role of motivation and the moderating effect of service time.

Mediation and moderation analyses were conducted with [15]’s ([15]) PROCESS macro for SPSS, the international benchmark for estimating conditional process models via regression. PROCESS generates bootstrap confidence intervals for indirect effects without assuming normality, incorporates interaction terms to test moderation, and conveniently combines both phenomena in moderated-mediation designs (e.g., model 14). Its user-friendly syntax and library of more than 90 predefined templates make it a robust and replicable choice for studies such as ours, which aim to determine whether the resilience–stress link is channeled through motivation and varies by teaching tenure.

The analysis in Table 3 shows a significant and positive relationship between resilience and motivation, evidenced by a coefficient of 0.413 and a *p*-value < 0.001, which reinforces the importance of resilience as a key explanatory factor in teacher motivation. The model shows that resilience contributes 35% to teacher motivation, suggesting that those with higher levels of resilience may have a greater willingness to face work challenges, positively influencing their motivation. This finding highlights the relevance of fostering resilience in teachers, as it could serve as a vehicle to improve their performance and well-being, especially in highly demanding contexts.

On the other hand, Table 4 presents more nuanced results on the influence of variables on job stress. In this case, it is observed that motivation has a significant negative effect on job stress (*p* < 0.0401), suggesting that higher levels of motivation could be associated with a lower perception of stress, which is consistent with psychological theories that indicate that motivation can act as a buffer against stressors in the work environment. This finding highlights the importance of cultivating an environment that promotes intrinsic motivation in teachers, as it could play a crucial role in mitigating their job stress. However, resilience did not show a significant effect on job stress (*p* > 0.5420), which could indicate that, although resilience is an important factor in coping with difficulties, it alone is not sufficient to reduce teachers’ job stress in this context. Similarly, length of service also did not present a significant effect (*p* > 0.9003), suggesting that accumulated experience, although valuable, is not a determining factor in stress reduction, at least not in terms of its direct impact in this study (Figure 2).

Finally, the interaction between resilience and length of service was also not significant (*p* > 0.8144), reinforcing the idea that resilience does not interact moderately with work experience to influence stress. This finding invites a questioning of the assumption that more experienced teachers necessarily develop a greater capacity to manage stress, suggesting that other factors, such as institutional support or job satisfaction, may play a more prominent role in this process.

The methods used to determine stress, resilience, and motivation levels in this study provide important context for interpreting the results. After collecting data using the standardized instruments, the researchers employed percentile-based categorization to establish meaningful threshold levels for each variable.

Specifically, scores were classified into three categories using percentile cutoffs: low (bottom 25%), moderate (25th to 75th percentile), and high (above 75th percentile). This approach allowed for a normative interpretation of scores relative to the study population rather than relying on arbitrary numerical thresholds. Using percentile-based categorization ensured that the distribution of participants across categories reflected the actual distribution of the measured attributes within the Chepén teacher population.

Resilience was measured using a 25-item questionnaire adapted from [14] ([14]), capturing five dimensions. The percentile-based categorization revealed that 11.1% of teachers fell into the low resilience category (below the 25th percentile), 51.1% exhibited moderate resilience (between 25th and 75th percentiles), and 37.8% demonstrated high resilience (above the 75th percentile). This distribution suggests that while most teachers possess adequate resilient capacities, a significant minority may require targeted support to develop these skills further (Table 5).

Work stress was assessed using [22]’s ([22]) 22-item questionnaire. Using the same percentile cutoffs, 13.3% of teachers experienced low stress levels, 42.2% reported moderate stress, and 44.5% indicated high to very high stress levels. These findings highlight the considerable occupational pressure experienced by nearly half of the teacher population, underlining the urgency of effective interventions.

Similarly, motivation was evaluated through [5]’ ([5]) 20-item questionnaire. The percentile-based analysis showed that 15.6% of teachers had low motivation (below 25th percentile), 37.8% demonstrated moderate motivation (25th–75th percentile), and 46.6% exhibited high to very high motivation (above 75th percentile). This distribution reveals a generally motivated teaching workforce, though the presence of a less motivated segment warrants attention.

This percentile-based approach to categorization provides a more contextually relevant interpretation of scores than absolute thresholds, accounting for the specific characteristics of the Chepén educational environment. It allows for meaningful comparisons within the study population while providing clear benchmarks for identifying teachers who might benefit most from targeted interventions.

### Explaining the Indirect Relationship Between Resilience and Work Stress

The findings reveal an interesting pattern regarding the relationship between resilience and work stress. While resilience significantly predicted motivation (β = 0.413, *p* < 0.001), and motivation in turn was significantly associated with reduced work stress (β = 0.335, *p* = 0.0401), resilience itself did not demonstrate a direct effect on work stress (β = 0.187, *p* = 0.5420). This finding appears to contradict some previous studies which suggested a direct protective effect of resilience against stress.

Several explanations may account for this apparent contradiction. The contextual specificity of teaching in Chepén likely plays a critical role, as educators in this region face unique challenges, including resource limitations and post-pandemic adaptations, that may alter how resilience functions in relation to stress. Additionally, the measurement approach used in this study captures different dimensions of resilience and stress compared to previous research, potentially representing more nuanced constructs.

The results suggest that resilience alone may be necessary but insufficient to reduce work stress in educational settings. While resilient teachers possess greater psychological resources, these resources must be channeled through motivational processes to effectively reduce stress reactions. This aligns with complex psychological models that propose motivation as an essential component in the stress-coping process, serving as the mechanism that activates resilient capacities toward stress management.

The high-stress educational environment of the post-pandemic period may have also temporarily overwhelmed the direct protective effects of resilience for many teachers. Under such circumstances, even highly resilient individuals require additional motivational resources to effectively manage stress, explaining why the resilience–stress relationship operates primarily through motivational pathways in this study.

These findings contribute important nuance to the understanding of resilience in occupational well-being, suggesting that its role may be more complex than previously thought, particularly in challenging educational contexts. 

## 5. Discussion

The data collected in this research, the objective of which is to study the relationship between resilience, work stress and motivational factors among teachers in a school in Chepén in 2024, yielded significant statistical results that contributed to a more solid understanding of the relationships between the factors that affect the psychological and professional well-being of school teachers. The first finding that was verified is the mediating effect of motivation on the relationship between resilience and school stress. Mediation was verified with a *p*-value of 0.001, which indicates that resilience affects motivation more strongly and motivation affects school stress (*p* < 0.0401). This finding is significant in terms of implying that resilience alone does not affect school stress and that this effect is realized through motivation. Therefore, teachers with higher resilience have lower school stress, but only when motivation is a mediator of this relationship. The contribution of motivation to job stress, measured with R^2^ = 20, also highlights the strong influence of emotional factors on teachers’ ability to withstand stress. This finding is similar to [10]’s ([10]) thesis on the mediating effect of motivation on the relationship between resilience and stress and helps to verify the hypothesis. Berdida’s scattered disagreement on motivation also emphasizes the weakness of service time, which in this study also had even lower significance. This may indicate that teachers tend to face stress challenges no matter how long they have worked. Regarding resilience levels, 11.1% of teachers present a low level of resilience, while 51.1% remain at a medium level. The study confirms previous evidence in this regard, such as [3] ([3]). The observed prevalence of moderate resilience also alerts us to the need to foster strategies that increase resilience in our teachers. The concept of resilience has been consistent with the interest there has been in the educational field on stress management and motivation. Only 13.3% of the teaching staff reported low stress, whereas 42.2% showed moderate stress, 33.3% high stress and 11.2% very high stress. In addition, the motivation level of 37.8% of the teachers was medium, and 26.6% reported a high level. This finding for the moderate level of motivation is complementary to the research of [2] ([2]), who showed that resilience could be directly associated with motivation—once the teacher is more resilient, that teacher is motivated. However, previous studies have shown that motivation is even higher in teachers. Overall, our findings are essential and demonstrate that resilience and motivation are interdependent. Therefore, these findings show the intricate web of evolutionary interactions between resilience and motivation. To address job stress in the teacher, factors such as resilience and motivation must be considered. Since mitigating job stress in the teacher is important, these factors improve not only the ability to cope with stressful incidents, but also increase overall well-being and productivity in the work environment.

The study achieved its primary objectives by confirming the mediating role of motivation between resilience and work stress, and by detailing the significant relationships between resilience and motivation, and motivation and stress. The non-significant findings regarding years of service as a moderator, rather than detracting from the study, highlight the complexity of this relationship and suggest that other factors may be more influential. Moreover, while the study does not offer specific practical recommendations, the detailed analysis of the relationships between resilience, motivation, and stress provides a strong foundation for the development of targeted interventions in future work.

The non-significant moderating effect of years of service on the resilience–stress link warrants deeper consideration. Recent studies have likewise reported null or inconsistent findings, suggesting that professional tenure does not uniformly buffer teacher stress. One plausible explanation is range restriction: in our sample, nearly ninety per cent of participants cluster between five and twenty years of service, limiting the statistical variability required to detect interactions. Second, collinearity with age may weaken the power of the interaction term because tenure and age typically correlate above 0.80. Third, the relationship could be curvilinear; longitudinal research describes U-shaped trajectories—higher stress at very early and very late career stages—that are obscured when only the linear term is modelled. Moreover, the Job Demands–Resources framework posits that organizational resources (e.g., principal support, collegial climate) can outweigh demographic factors, and motivation, which in our model played a robust mediating role, may absorb variance otherwise attributable to tenure. To probe further, we recommend (a) adding the quadratic term for years of service and its interaction with resilience, (b) conducting a Johnson–Neyman analysis to identify tenure ranges where resilience does buffer stress, and (c) incorporating personal and organizational resources—teacher self-efficacy, psychological capital, administrative support—as potential competitive moderators. Unpacking these nuances will enable tailored interventions: mentoring for novices, vocational renewal programs for veterans, and workload adjustments aligned with career stage.

## 6. Conclusions

Thus far, the results of this study confirm the alternative hypothesis, demonstrating that the findings indicate that resilience relates to lower work stress only indirectly, via the increase in motivation; the direct effect of resilience on stress was not statistically significant. The use of the statistical tool PROCESS v4.3 for SPSS has shown that resilience has a direct impact on motivation, being the responsible source for 35% of its behavior (R-sq = 35%). Likewise, motivation has a significant impact on job stress, with an R-sq of 20% and a *p*-value < 0.0401, indicating that motivation plays a crucial role in reducing job stress. This finding clarifies that resilience, in itself, does not directly influence stress, but its impact is understood only when motivation mediates this relationship. Therefore, job stress decreases only in combination with high levels of resilience when motivation is also high. Consequently, this finding implies that occupational health is ensured only when both parameters are high. Looking at resilience, the analysis of the results indicates the presence of level 3 average resilience among the majority of teachers, with 51.1% of the values. This finding could mean that, although there is some coping ability, such ability is not sufficient to provide an optimal response to the emotions and pressures central to the teaching environment. In addition, 11.1% of teachers are at level 1, suggesting that their vulnerability is more significant, and 17.8% and 20% are level 4 and 5 values, respectively. This implies that there is a significant segment of the teaching staff that can thrive through adversities. In terms of job stress levels, the results suggest an alarming reality: 33.3% of teachers report high levels of stress, and 11.2% present very high levels, emphasizing the significant prevalence of this problem in the educational setting. Although 42.2% of teachers are at medium levels, which could be interpreted as less intense expert pressure, high levels of stress are still significant and could be detrimental to emotional well-being and work efficiency. This relationship indicates an urgent need for interventions that support stress reduction among teachers, and resilience and motivation enhancement approaches can be considered as such. In the case of motivation, the results reveal that 15.6% of teachers have low levels of motivation. It is possible that this figure is related to stress and professional dissatisfaction. However, 37.8% are at a medium level and could be considered moderately motivated, and 26.6% have high levels of motivation, while 20% have very high levels. This fact shows that a significant proportion of teachers maintain high levels of interest in their work. Therefore, the development of a high-motivation environment should be encouraged, especially for those with low motivation. Improving working conditions, development opportunities, and work collaboration could be an appropriate approach to increase motivation and thereby decrease the prevalence of job stress. The results confirm that resilience is linked to lower work stress only through its positive effect on motivation; the direct effect of resilience on stress was not significant. Overall, the results clearly indicate the relationship between resilience and motivation and decreased stress among teachers. As such, educational institutions should focus on fostering these qualities, as this could benefit not only the well-being of teachers, but perhaps the quality of education and long-term viability

### 6.1. Limitations and Future Research

Several limitations warrant consideration. The cross-sectional design precludes firm causal inferences regarding the links among resilience, motivation and stress. All variables were assessed through self-report instruments, exposing the data to potential social-desirability bias and common-method variance. Although we conducted a census, participants came from a single district, limiting the extent to which findings can be generalized to other cultural or educational settings. Confidentiality procedures likely reduced but did not entirely eliminate recall bias. Future research should employ longitudinal or intervention designs to trace motivational and stress trajectories over time and to test the proposed causal chain more rigorously. Complementing self-reports with objective indicators or third-party evaluations, replicating the model in diverse regions and using multilevel analyses to capture organizational influences would also enhance external validity

### 6.2. Specific Intervention Recommendations

Based on the findings that resilience influences work stress primarily through motivation, several specific interventions are recommended:Resilience-Building Workshops: Implement structured 8–10 session workshops focusing specifically on the resilience dimensions found to be most predictive of motivation. These should include practical exercises in cognitive reframing, mindfulness practices, and problem-solving strategies tailored to educational challenges in Chepén schools.Motivation Enhancement Programs: Develop targeted interventions that address both intrinsic and extrinsic motivation factors. For intrinsic motivation, create professional learning communities where teachers can explore personal growth opportunities and reconnect with their teaching purpose. For extrinsic motivation, establish recognition systems and career advancement pathways that acknowledge teacher achievements.Integrated Well-being Approach: Rather than addressing resilience or stress in isolation, implement comprehensive well-being programs that explicitly connect resilience-building activities with motivational enhancement. These could include mentoring relationships between experienced and newer teachers to share coping strategies while fostering professional motivation.Organizational-Level Changes: Beyond individual interventions, implement structural changes such as redistributing administrative tasks, creating dedicated planning time, and establishing clear communication channels to address organizational stressors identified in the research.Customized Interventions by Experience Level: Since years of service did not moderate the resilience–stress relationship as expected, develop differentiated support systems that address the specific motivational needs of teachers at different career stages. Early-career teachers might benefit from competence-building support, while veteran teachers might need interventions focused on renewing purpose and preventing burnout.Psychological Resource Training: Implement specific training in psychological resource management that explicitly bridges resilience capacities with motivational orientation. This could include workshops on identifying personal and professional values, aligning daily practices with these values, and cultivating a sense of purpose that enhances motivation even during challenging periods.

These findings and recommendations are consistent with [10]’s ([10]) research on the complementary relationship between flexibility and motivation, while extending previous work to specifically identify the mediating mechanisms. The study also supports [2]’s ([2]) findings on the direct association between resilience and motivation, though our research further elucidates how this relationship impacts work stress in educational settings.

This research demonstrates that occupational well-being in educational settings requires attention to both resilience-building and motivational enhancement. Interventions that target only one factor are likely to be insufficient for reducing work stress effectively. Educational leaders and policymakers should prioritize programs that strengthen the resilience-motivation pathway to create healthier, more productive teaching environments in Chepén and similar educational contexts

## Figures and Tables

**Figure 1 behavsci-15-00888-f001:**
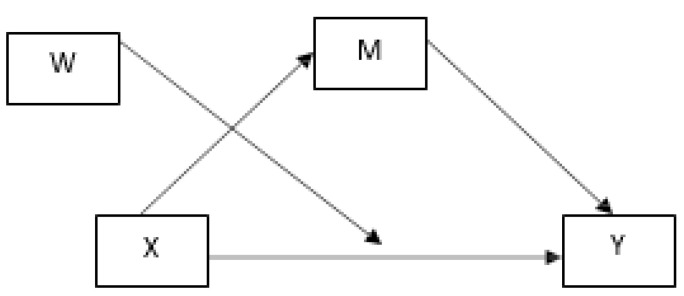
Outline of multivariate research design. X: Resilience, Y: Work stress, M: Motivation, W: Time of service.

**Figure 2 behavsci-15-00888-f002:**
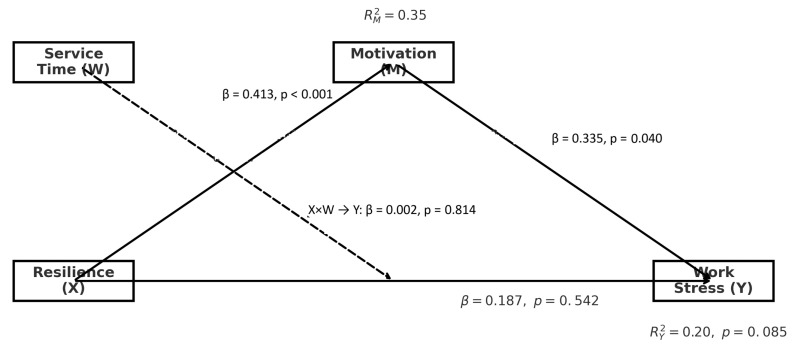
Solved conceptual model.

**Table 1 behavsci-15-00888-t001:** Sociodemographic characteristics of participants (*n* = 450).

Variable		%
Sex	Female	75.6
	Male	24.4
Age (years)	Under 30	11.1
	30 to 45	31.1
	45 and over	57.8
Service time (years)	Less than 5	11.1
	5 to 20	40.0
	20+	48.9

**Table 2 behavsci-15-00888-t002:** Instrument reliability analysis (*n* = 450).

Variable	N° of Items	Cronbach’s Alpha	Reliability Level
Resilience	25	0.859	Good
Motivation	20	0.870	Good
Work Stress	22	0.716	Acceptable

**Table 3 behavsci-15-00888-t003:** Effect on the mediating variable (first stage).

Variable	Coef.	SE	t	*p*	95% CI
Resilience	0.413	0.093	4.439	0.0001	[0.225, 0.601]
R^2^ = 0.35, F (1,42) = 19.7118, *p* = 0.0001					

**Table 4 behavsci-15-00888-t004:** Effect on the dependent variable (second stage).

Variable	Coef.	SE	t	*p*	95% CI
Resilience	0.187	0.304	0.615	0.5420	[−0.428, 0.802]
Motivation	0.335	0.158	2.123	0.0401	[−0.655, −0.016]
Service time	0.217	1.727	0.126	0.9003	[−3.276, 3.712]
Interaction	0.002	0.011	0.236	0.8144	[−0.026, 0.021]
R^2^ = 0.20, *p* = 0.085					

**Table 5 behavsci-15-00888-t005:** Level of resilience, work stress and motivation.

Level	Resilience	Work Stress	Motivation
Low	11.1%	13.3%	15.6%
Medium	51.1%	42.2%	37.8%
High	17.8%	33.3%	26.6%
Very high	20.0%	11.2%	20.0%

## Data Availability

Data are contained within the article.

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
