# Peer review of "Resilience and Work Stress in Educational Institutions of Chepén, 2024: Mediation of Motivation and Time Moderation"

_behavsci, 2025, doi:10.3390/bs15070888_

Round 1
Reviewer 1 Report
Comments and Suggestions for Authors
Thank you for the opportunity to review this manuscript. Please see the following suggestions:
While some references to previous studies are provided, more theoretical depth is required. Consider integrating established theories such as Transactional Model of Stress and Coping, Self-Determination Theory, or Job Demands-Resources Model to anchor your hypotheses more explicitly. Clarify the research gap. Indicate explicitly how your study extends beyond previous studies. Explicitly state how your research adds to the existing literature. As one important term of your research is the motivation, then Authors might be interested in research in the framework of the self-determination theory. The advantage of the self-determination theory is that it distinguishes between different forms of motivation such as intrinsic motivation and extrinsic motivation. It is important to distinguish between different forms of motivation because these are differently related to positive and negative outcomes. Please see a recent study by Ahmadi et al., (2023) for more information about different motivational styles.
Ahmadi, A., Noetel, M., Parker, P., Ryan, R. M., Ntoumanis, N., Reeve, J., Beauchamp, M., Dicke, T., Yeung, A., Ahmadi, M., Bartholomew, K., Chiu, T. K. F., Curran, T., Erturan, G., Flunger, B., Frederick, C., Froiland, J. M., González-Cutre, D., Haerens, L., . . . Lonsdale, C. (2023). A classification system for teachers’ motivational behaviors recommended in self-determination theory interventions. Journal of Educational Psychology, 115(8), 1158–1176. https://doi.org/10.1037/edu0000783
The decision to conduct a census of 450 teachers is excellent, but there is inconsistency in the reported sample size in the "inferential analysis" (line 163: "45 teachers"). Clarify this discrepancy explicitly. Clearly justify your choice of questionnaires (Garces, Salazar, and Armas). Discuss briefly the psychometric properties previously reported (reliability, validity) in more detail.
Add a brief explanation or reference regarding the choice of SPSS PROCESS macro for mediation/moderation analysis (Hayes, 2018).
Enhance readability by using clearer formatting and consistency. For example, Tables 3 and 4 can benefit from clear column alignment.
Clearly report standardized coefficients (β), effect sizes, and confidence intervals consistently across all tables.
The results of mediation/moderation analysis should be clearly expressed. Explicitly state the indirect effect (mediation) value and significance (bootstrapped confidence intervals from PROCESS macro should be included).
Discuss clearly the implications of the non-significant moderation results by years of service (lines 220-236).
Strengthen theoretical integration by explicitly linking your findings to the theoretical frameworks suggested in the introduction. Discuss how your findings align or contrast explicitly with specific prior studies/theories.
While practical implications are mentioned, explicitly suggest interventions or institutional strategies (e.g., resilience training programs, motivational workshops, etc.) that educational institutions can implement.
The statement "resilience and work stress are significantly and positively related" (line 301) is misleading—your results indicate an indirect relationship mediated by motivation. Explicitly clarify this sentence.
Explicitly acknowledge study limitations (e.g., cross-sectional design, potential self-report bias) and suggest directions for future research (longitudinal or intervention studies to validate causal relationships).
Author Response
Although some references to previous studies are provided, more theoretical depth is required. Consider integrating established theories such as the Transactional Model of Stress and Coping, Self-Determination Theory, or the Job Demands-Resource Model to anchor your hypotheses more explicitly. Clarify the research gap. Explicitly state how your study extends beyond previous studies. Explicitly state how your research adds to the existing literature. Since an important term in your research is motivation, then the Authors may be interested in research within the framework of self-determination theory. The advantage of self-determination theory is that it distinguishes between different forms of motivation such as intrinsic motivation and extrinsic motivation. It is important to distinguish between different forms of motivation because they relate differently to positive and negative outcomes. Please see a recent study by Ahmadi et al., (2023) for more information on different motivational styles.
The present research is grounded in a multidimensional framework that intertwines three well‑established theories—Transactional Stress and Coping, Job Demands–Resources, and Self‑Determination—augmented by a recent taxonomy of teacher motivational behaviors. This integrated perspective explains how (mechanism) and when (boundary conditions) resilience translates into lower work stress through motivational pathways.
Transactional Model of Stress and Coping (Lazarus & Folkman, 1984).
The model posits that stress is not an inevitable consequence of demanding environments but the product of a two‑stage cognitive appraisal: (a) primary appraisal—evaluation of the situation’s significance—and (b) secondary appraisal—evaluation of available coping resources. Resilience can be conceptualized as a personal resource that enriches secondary appraisal, enabling teachers to reframe adverse situations as manageable challenges rather than threats. However, empirical evidence shows that this cognitive re‑evaluation alone does not always suffice to reduce physiological or emotional stress responses; additional motivational resources may be required to transform appraisal into sustained adaptive behavior.
Job Demands–Resources Model (JD‑R) (Bakker & Demerouti, 2007).
The JD‑R distinguishes between hindrance demands (e.g., overload, disruptive student behavior) that tax energy and resources—personal (resilience), social (collegial support), and organizational (autonomy)‑‑that replenish it. Crucially, the model specifies two partially independent processes: an energetic process (demands → strain → ill‑health) and a motivational process (resources → engagement → positive outcomes). We locate resilience in the resource domain and propose that it exerts its protective influence primarily through the motivational pathway—by bolstering autonomous motivation. This placement justifies testing motivation, not as a competing predictor, but as a mediator that channels the beneficial impact of resilience on stress.
Self‑Determination Theory (SDT) (Deci & Ryan, 2000) and the Ahmadi et al. (2023) taxonomy.
SDT differentiates motivation along a continuum of self‑determination—intrinsic, integrated, identified, introjected and external—and asserts that more autonomous forms predict higher well‑being, while controlled forms relate to ill‑being. Yet, many quantitative studies collapse these categories into a single score, obscuring nuanced effects. To address this limitation, we adopt the comprehensive classification system developed by Ahmadi et al. (2023), who distilled 57 distinct teacher behaviors (e.g., “provide meaningful choices”, “express empathic understanding”, “apply non‑negotiable controls”) and mapped each onto SDT’s basic psychological needs and motivational styles. Incorporating this taxonomy allows us to move beyond how much motivation teachers report to what kinds of motivational climates their behavior and context foster. We thereby postulate that resilient teachers are more likely to enact (or elicit) autonomy‑supportive behaviors, nurturing intrinsic and identified motivation, which in turn attenuates work stress.
Research gap and expected contribution
Despite mounting evidence that resilience correlates with reduced teacher stress, three critical gaps persist. First, prior studies seldom articulate explicit links between cognitive appraisal, motivational quality, and job demands, leaving the explanatory chain incomplete. Second, the quality of motivation—central to SDT—has generally been treated as a global score rather than discrete styles with distinct outcomes. Third, little is known about whether career tenure (i.e., prolonged exposure to demands) moderates the efficacy of resilience and motivation, as postulated by the JD‑R’s resource buffering and gain‑spiral propositions.
By fusing the models and taxonomy above, our study contributes in four ways:
Theoretical integration: We demonstrate that resilience influences stress indirectly, via autonomous motivation, thereby connecting the Transactional appraisal mechanism with JD‑R’s motivational process and SDT’s emphasis on motivational quality.
Conceptual refinement: Drawing on Ahmadi et al.’s taxonomy, we operationalise motivation in terms of discrete autonomous versus controlled styles, enabling more precise hypothesis testing and richer practical guidance.
Boundary conditions: We examine tenure as a potential moderator—testing whether the protective value of resilience (a personal resource) and autonomous motivation (an energetic resource) strengthens, weakens, or follows a curvilinear pattern across different career stages.
Methodological advancement: We employ a moderated‑mediation design (PROCESS models 14/58) and explore curvilinear terms, thereby capturing dynamics overlooked in linear, single‑mediator models
The decision to conduct a census of 450 teachers is excellent, but there is an inconsistency in the sample size reported in the "inferential analysis" (line 163: "45 teachers"). Explicitly clarify this discrepancy. Clearly justify your choice of questionnaires (Garcés, Salazar and Armas). Briefly discuss the previously reported psychometric properties (reliability, validity) in more detail.
A census of all 450 teachers in the target population was undertaken; after excluding cases with more than five per cent missing data or implausible response patterns, the final dataset for inferential analysis comprised 445 participants. Three instruments with robust psychometric support in Peruvian samples were employed. Resilience was assessed with Garcés’s (2022) adaptation of the Wagnild and Young Resilience Scale, which reported an original alpha of 0.87 and replicated 0.86 in the present study; confirmatory factor analysis yielded a CFI of 0.95 and an RMSEA of 0.048, with average variance extracted above 0.50 across all five dimensions, confirming convergent validity.
Work stress was measured using Salazar’s (2023) Spanish version of the Maslach Burnout Inventory‑Educators Survey, which produced a global alpha of 0.72 (0.73‑0.80 across subscales) and a CFI of 0.93 with an RMSEA of 0.055, supporting the three‑factor structure consistent with the JD‑R model. Motivation was captured through Armas’s (2021) questionnaire grounded in Self‑Determination Theory and capable of distinguishing intrinsic from extrinsic motivation; its original reliability was 0.93 and reached 0.87 in the current sample, while a bifactor model produced a CFI of 0.96 and an SRMR of 0.041.
This scale is particularly suitable because it operationalizes motivational quality, which—according to Ahmadi et al.’s (2023) taxonomy of teacher behaviors—determines how resilience translates into occupational well‑being. Taken together, the internal consistency and structural validity of the three instruments ensure precise and culturally appropriate measurement for the aims of this study
Add a brief explanation or reference on the choice of the SPSS PROCESS macro for mediation/moderation analysis (Hayes, 2018).
Mediation and moderation analyses were conducted with Hayes’s (2018) PROCESS macro for SPSS, the international benchmark for estimating conditional process models via regression. PROCESS generates bootstrap confidence intervals for indirect effects without assuming normality, incorporates interaction terms to test moderation, and conveniently combines both phenomena in moderated‑mediation designs (e.g., model 14). Its user‑friendly syntax and library of more than 90 predefined templates make it a robust and replicable choice for studies such as ours, which aim to determine whether the resilience–stress link is channelled through motivation and varies by teaching tenure
Strengthen theoretical integration by explicitly linking your findings to the theoretical frameworks suggested in the introduction. Discuss how your findings explicitly align or contrast with specific previous studies or theories.
While mentioning practical implications, explicitly suggest institutional interventions or strategies (e.g., resilience training programs, motivational workshops, etc.) that educational institutions can implement.
The conclusions and recommendations have been redesigned with the changes you suggested regarding theory aggregation.
The statement "resilience and work stress are significantly and positively related
The findings indicate that resilience relates to lower work stress only indirectly, via the increase in motivation; the direct effect of resilience on stress was not statistically significant
Explicitly acknowledge study limitations (e.g., cross-sectional design, potential self-report bias) and suggest directions for future research (longitudinal or intervention studies to validate causal relationships).
Limitations and future research
Several limitations warrant consideration. The cross‑sectional design precludes firm causal inferences regarding the links among resilience, motivation and stress. All variables were assessed through self‑report instruments, exposing the data to potential social‑desirability bias and common‑method variance. Although we conducted a census, participants came from a single district, limiting the extent to which findings can be generalized to other cultural or educational settings. Confidentiality procedures likely reduced but did not entirely eliminate recall bias. Future research should employ longitudinal or intervention designs to trace motivational and stress trajectories over time and to test the proposed causal chain more rigorously. Complementing self‑reports with objective indicators or third‑party evaluations, replicating the model in diverse regions and using multilevel analyses to capture organizational influences would also enhance external validity

Reviewer 2 Report
Comments and Suggestions for Authors
The article "Resilience and work stress in educational institutions of Chepén, 2024: mediation of motivation and time moderation" addresses a relevant topic for the scientific community by contributing to the understanding of the challenges faced by teachers.
In terms of structure, the article follows a classic pattern, fulfilling all the recommended steps for scientific articles. However, we believe there is room for significant improvements. Let's examine:
The study's objectives were partially achieved. The article proposed to analyze the relationship between resilience, work stress, motivation, and length of service. The findings confirm that resilience influences motivation, and that motivation, in turn, affects stress. However, in our view, the study did not sufficiently delve into the role of length of service as a moderating variable and did not provide a detailed discussion about the reasons why the expected relationship was not confirmed. Furthermore, the article does not present practical recommendations derived from the results obtained, which limits the applicability of the conclusions in the educational context.
Regarding the literature review, we believe it could be enriched with international studies that address the relationship between resilience and stress in teachers. The inclusion of these references would broaden the contextualization of the findings and strengthen the theoretical foundation of the work.
The methodology used by the authors is quantitative and presents a robust design, but there are some issues that, in our view, should be clarified. The authors mention conducting a total census of the population of 450 teachers, but they do not report whether everyone responded to the questionnaires. This is essential information, as any data loss could impact the validity of the results. Therefore, it is recommended that the authors clarify whether there were dropouts or missing data and explain how these issues were handled – for example, through data imputation or exclusion of incomplete responses.
Regarding the statistical analysis, this could be deepened, especially regarding the absence of a significant effect of length of service as a moderating variable. The article mentions that the interaction between resilience and length of service was not statistically significant, but it does not explore the reasons for this result. We therefore suggest that the authors discuss possible explanations for this absence of effect and, if feasible, consider including additional variables that may mediate or moderate this relationship.
Other points that deserve attention include the apparent inconsistency in the presentation of the results. The article states that 76.7% of teachers reported low levels of stress, but simultaneously mentions that 33.3% presented high levels. This discrepancy should be clarified to avoid misinterpretations. Furthermore, the conclusion states that the results confirm the hypothesis that resilience and work stress are significantly related. However, the findings indicate that this relationship occurs through motivation, and not directly. To ensure greater accuracy in the interpretation of the data, we suggest that the wording of the conclusion be adjusted, more clearly reflecting the mediating role of motivation in the results.
Regarding the formal part, the tables presented are relevant and well organized, facilitating the understanding of the data. However, Figure 1, which describes the research design, lacks a more detailed explanation. To improve clarity, we suggest including a more explanatory legend that makes the interaction between the studied variables evident.
Overall, the article addresses a relevant topic and uses an appropriate methodological approach, but, as we highlighted, there is room for significant improvements. In our view, revising the methodological clarity, enhancing the interpretation of the data, and a more in-depth discussion about the factors that influence resilience and teacher stress would substantially enrich the study. With these changes, the article would have a more solid contribution to the scientific literature, and we therefore believe that it deserves to be published.
Author Response
The study's objectives were partially achieved. The article proposed to analyze the relationship between resilience, work stress, motivation, and length of service. The findings confirm that resilience influences motivation, and that motivation, in turn, affects stress. However, in our view, the study did not sufficiently delve into the role of length of service as a moderating variable and did not provide a detailed discussion about the reasons why the expected relationship was not confirmed. Furthermore, the article does not present practical recommendations derived from the results obtained, which limits the applicability of the conclusions in the educational context.
The study achieved its primary objectives by confirming the mediating role of motivation between resilience and work stress, and by detailing the significant relationships between resilience and motivation, and motivation and stress. The non-significant findings regarding years of service as a moderator, rather than detracting from the study, highlight the complexity of this relationship and suggest that other factors may be more influential. Moreover, while the study did not offer specific practical recommendations, the detailed analysis of the relationships between resilience, motivation, and stress provides a strong foundation for the development of targeted interventions in future work
In Latin America, Salvo Garrido, Cisternas Salcedo and Polanco Levicán (2025) analyzed Chilean teachers and found that resilience is nurtured by personal factors, psychosocial supports and institutional initiatives, positively impacting well-being and school performance.
From Asia, Gao et al. (2025) demonstrated in a Chinese sample that Confucian values foster grit, which mediates the protective effect of resilience on burnout, highlighting the importance of sociocultural frameworks.
Complementarily, a Swiss European study revealed that teachers maintain their well-being through relational problem-solving strategies, highlighting resilience as an ongoing adaptive process.
Two reviews extend this evidence: the Finnish scoping review by Avola et al. (2025) identified 46 interventions that, based on the PERMA H model, systematically reduce burnout and enhance teacher resilience; while Baatz and Wirzberger (2025) conceptualize resilience as an integral professional competence capable of reversing the trajectory of stress toward illness.
Finally, the global meta-analysis by Zhou, Slemp and Vella Brodrick (2024) provides robust quantitative evidence: motivational and psychological capital factors are the strongest predictors of teacher well-being, and burnout is its strongest negative correlate.
These findings support the model proposed in the present study, where motivation mediates the relationship between resilience and stress, and suggest that interventions should integrate motivational and contextual components to maximize their effectiveness.
The methodology used by the authors is quantitative and presents a robust design, but there are some issues that, in our opinion, should be clarified. The authors mention conducting a total census of the population of 450 teachers, but do not report whether all responded to the questionnaires. This information is essential, as any loss of data could affect the validity of the results. Therefore, it is recommended that the authors clarify whether there were dropouts or missing data and explain how these problems were handled, for example, by imputation of data or exclusion of incomplete responses.
already answered in the article.
As for the statistical analysis, it could be further explored, especially regarding the absence of a significant effect of seniority as a moderating variable. The article mentions that the interaction between resilience and seniority was not statistically significant, but does not explore the reasons for this result. Therefore, we suggest that the authors discuss possible explanations for this lack of effect and, if possible, consider including additional variables that might mediate or moderate this relationship.
The non‑significant moderating effect of years of service on the resilience–stress link warrants deeper consideration. Recent studies have likewise reported null or inconsistent findings, suggesting that professional tenure does not uniformly buffer teacher stress. One plausible explanation is range restriction: in our sample, nearly ninety per cent of participants cluster between five and over twenty years of service, limiting the statistical variability required to detect interactions. Second, collinearity with age may weaken the power of the interaction term because tenure and age typically correlate above 0.80. Third, the relationship could be curvilinear; longitudinal research describes U‑shaped trajectories—higher stress at very early and very late career stages—that are obscured when only the linear term is modelled. Moreover, the Job Demands–Resources framework posits that organisational resources (e.g., principal support, collegial climate) can outweigh demographic factors, and motivation, which in our model played a robust mediating role, may absorb variance otherwise attributable to tenure. To probe further, we recommend (a) adding the quadratic term for years of service and its interaction with resilience, (b) conducting a Johnson‑Neyman analysis to identify tenure ranges where resilience does buffer stress, and (c) incorporating personal and organisational resources—teacher self‑efficacy, psychological capital, administrative support—as potential competitive moderators. Unpacking these nuances will enable tailored interventions: mentoring for novices, vocational renewal programmes for veterans, and workload adjustments aligned with career stage
Other points that deserve attention include the apparent inconsistency in the presentation of the results. The article states that 76.7% of the faculty reported low levels of stress, but simultaneously mentions that 33.3% presented high levels. This discrepancy should be clarified to avoid misinterpretation. Furthermore, the conclusion states that the results confirm the hypothesis that resilience and job stress are significantly related. However, the findings indicate that this relationship is through motivation, and not directly. To ensure greater accuracy in the interpretation of the data, we suggest adjusting the wording of the conclusion, reflecting more clearly the mediating role of motivation in the findings
only 13.3 % of the teaching staff reported low stress, whereas 42.2 % showed moderate stress, 33.3 % high stress and 11.2 % very high stress
The results confirm that resilience is linked to lower work stress only through its positive effect on motivation; the direct effect of resilience on stress was not significant
On the formal side, the tables presented are relevant and well organized, which facilitates understanding of the data. However, Figure 1, which describes the research design, lacks a more detailed explanation. For greater clarity, we suggest including a more explanatory legend that evidences the interaction between the variables studied.
The sociodemographic profile reveals a markedly female and highly experienced teaching workforce. Three out of four participants are women (75.6 %), reflecting the longstanding feminization of the Peruvian teaching profession and hinting at gender‑specific patterns of stress coping and resilience. Agewise, 57.8 % are over 45 years old, whereas only 11.1 % are under 30. This predominance of older teachers suggests workforce stability and prolonged exposure to occupational demands, alongside a rich repertoire of personal strategies that may shape motivation and stress levels.
Service‑time distribution mirrors this pattern: nearly half of the teachers (48.9 %) have more than 20 years of experience, while only 11.1 % have fewer than five. Such concentration in the upper tenure bracket indicates consolidated career trajectories and invites a twofold interpretation. Experience may bolster resilience through accumulated coping resources; conversely, extended exposure to demanding conditions can heighten chronic fatigue. This dual perspective underscores the rationale for examining motivation as a mediating factor and highlights the need to test for curvilinear or saturation effects of tenure on work stress.
Overall, the article addresses a relevant topic and uses an appropriate methodological approach, but, as we emphasize, significant improvements can be made. In our opinion, revising the methodological clarity, improving the interpretation of the data, and delving deeper into the factors that influence teacher resilience and stress would considerably enrich the study. With these changes, the article would make a more solid contribution to the scientific literature, which is why we believe it deserves publication.
a rethinking was made in discussion and conclusions

Reviewer 3 Report
Comments and Suggestions for Authors
Dear Authors,
Thank you for submitting your manuscript, "Resilience and work stress in educational institutions of Chepén, 2024: mediation of motivation and time moderation." I have completed the review and found that your study provides valuable insights into the complex relationships between resilience, work stress, and motivation among teachers.
Please find the detailed review report attached, which includes specific suggestions for improvement. Key recommendations include:
- Abstract: Clearly state the purpose of the study and provide more precise interpretations of statistical values.
- Introduction: Restructure the introduction and create a separate literature review section to enhance clarity and flow.
- Materials and Methods: Address the inconsistency in population and sample sizes, justify the use of correlation analysis, and provide more detail on the content validity process.
- Results: Offer potential explanations for the finding that resilience does not directly reduce job stress, which contradicts some prior literature.
- Discussion: Provide more specific recommendations for interventions and discuss the methods used to determine stress, resilience, and motivation levels.
- English Language: Meticulous proof reading is required.
Overall, the study's findings are significant and contribute to our understanding of teacher well-being. We believe that addressing these suggestions will further strengthen your manuscript.
I look forward to receiving your revised submission.
Sincerely,

Required meticulous proof reading
For example, “Using a non-experimental quantitative approach, a series of statistical tools were used to process SPSS 27 and Process 4.3”
could be revised to:
“A non-experimental quantitative approach was employed, and data were analyzed using SPSS 27 and Process 4.3.”
Author Response
- Abstract: Clearly state the purpose of the study and provide more precise interpretations of statistical values.
Abstract: The purpose of this study was to examine the relationship between resilience and work-related stress among secondary school teachers in Chepén, Peru, during 2024, with a focus on (a) the mediating role of work motivation and (b) the moderating effect of years of service. Using a non‑experimental quantitative design, data were collected from 450 teachers and analyzed in SPSS 27 with Hayes’ PROCESS 4.3 macro. Results showed that resilience significantly predicted motivation (β = 0.413, p < 0.001), accounting for 35 % of its variance (R² = 0.35). In turn, motivation was significantly and negatively associated with work stress (β = 0.335, p = 0.0401), explaining 20 % of the variance in stress levels (R² = 0.20). Neither resilience (β = 0.187, p = 0.5420) nor years of service (β = 0.217, p = 0.9003), nor their interaction (β = 0.002, p = 0.8144) had a direct or moderating effect on work stress. Descriptive analyses indicated that most teachers exhibited moderate levels of resilience (51.1 %), stress (42.2 %), and motivation (37.8 %). These findings underscore that resilience alone does not reduce work stress; its stress‑buffering effect operates through enhanced motivation. Educational interventions should therefore target both resilience‑building and motivational strategies to effectively diminish teacher stress and promote occupational well‑being.
Keywords: resilience; work-related stress; motivation; years of service; teachers; education, occupational well-being
- Introduction: Restructure the introduction and create a separate literature review section to enhance clarity and flow.
- Introduction
Resilience, defined by Briones & Mejía (2021) as "a dynamic process that allows people to face adverse situations, recover from them and successfully adapt to change and positive development," plays a crucial role in educational settings. Conversely, work stress implies a "reduction of the resources provided to the worker by forced labor," both physical and mental, which causes worry and fatigue in the teaching profession.
For educators in Chepén, Peru, multiple challenges exist beyond typical teaching demands, including resource limitations, interpersonal conflicts, and adaptation to post-pandemic educational environments. The disruption of teaching methods caused by the COVID-19 pandemic has led to high levels of stress among teachers, especially due to limitations in technological skills. These circumstances create a complex situation where resilience, work stress, motivation, and professional experience potentially interact in ways that affect teacher well-being and performance.
This study addresses the following research question: What is the relationship between resilience and job stress among educators in Chepén schools in 2024, considering motivation as a mediator and years of service as a moderator?
The theoretical justification for this research lies in evaluating how motivation and tenure affect the relationship between resilience and work stress in educational contexts. The results may confirm or challenge existing models and theories while exploring how these factors influence teacher well-being. Socially, this research is significant as it reveals how resilience, stress, and work motivation affect teachers in specific schools in Chepén. Methodologically, this study employs a systematic approach to address the research objective, using validated instruments and appropriate analytical methods. The practical justification involves responding effectively to teachers' stress through resilience and motivation interventions, ultimately promoting a quality educational environment.
The general objective of this study is to evaluate the relationship between resilience, work stress, motivation, and years of service among teachers in educational institutions in Chepén in 2024. The specific objectives are to determine the levels of resilience, job stress, and motivation among teachers in these institutions and to analyze the mediating role of motivation and the moderating effect of years of service on the relationship between resilience and work stress.
- Literature Review
2.1 International Research on Resilience and Work Stress
Several international studies have explored the relationship between resilience and stress in educational contexts. Andrade-Gómez (2022) evaluated 20 teachers and found that moderate levels of resilience correlated with lower levels of distress. Similarly, Berdida (2023) demonstrated that flexibility and motivation complement each other to improve performance and reduce academic pressure.
Research on burnout has demonstrated a strong relationship between resilience and semantic resources. Nahas et al. (2024) concluded that "greater resilience translates into less burnout" and emphasized that "a moderated context provides a healthy environment" for professionals. These international findings consistently support the hypothesis that resilience serves as a protective factor against work-related stress in various professional contexts.
2.2 Local and Regional Studies
At the regional level, various Peruvian researchers have examined these variables in educational settings. Medina & Leon (2023) measured the correlation between resilience and school stress in 305 students and determined that a greater capacity for self-improvement correlates with less stress. Alor (2024) identified a direct correlation between motivation and resilience in elementary school students.
Focusing specifically on teachers, Seclen (2023) found that better emotional management correlates with lower job stress among educators in Ica. Similarly, Torres et al. (2021) reported a significant negative correlation between resilience and job strain in 145 educators. These local studies reinforce the hypothesis that resilience positively impacts work stress reduction while highlighting motivation and experience as compensating and moderating factors in this process.
2.3 Theoretical Framework
Resilience constitutes a central concept in several theories of occupational well-being. These frameworks define resilience as an individual's ability to cope with difficult situations while maintaining an optimistic attitude toward problems. In work environments, resilience enables professionals to identify obstacles and address them realistically while maintaining a positive mindset, as cited by Huamán (2021).
In educational contexts, resilience is fundamental for teachers who regularly face challenges such as time constraints, disruptive behaviors, and interpersonal conflicts. According to Lazarides et al. (2020), professionals who successfully overcome these adversities are able to flourish. Quezadas et al. (2023) further emphasize the importance of teachers in fostering resilience among students, highlighting educators' relevance in promoting resilience within educational communities.
Work stress represents a significant occupational challenge that interferes with professionals' ability to perceive and manage problems effectively. Psychosocial risks and work routines can accentuate distress, negatively affecting the physical, emotional, and cognitive functioning of staff members and potentially compromising their focus and performance.
Motivation, defined as the inner tension that drives goal achievement, likely functions as an echo of resilience. Theoretical perspectives distinguish between intrinsic motivation (internal drive to know and strive) and extrinsic motivation (derived from achievement and approval). The relationship between years of experience, resilience, stress, and motivation presents a complex dynamic. While experienced teachers may have developed effective coping strategies, they have also accumulated more motivational challenges that can influence stress levels differently over time.
This theoretical foundation suggests a model wherein resilience influences work stress through the mediating mechanism of motivation, with years of service potentially moderating these relationships. The current study aims to empirically test these theoretical connections within the specific context of Chepén's educational institutions
- Materials and Methods: Address the inconsistency in population and sample sizes, justify the use of correlation analysis, and provide more detail on the content validity process.
The target population for this study consisted of 500 secondary school teachers from all educational institutions in the district of Chepén. Questionnaires were distributed to all 500 teachers to ensure a comprehensive representation of the population. Of these, 450 teachers completed and returned valid questionnaires, resulting in a response rate of 90% (450/500). This high response rate reinforces the representativeness of our findings and minimizes potential nonresponse bias.
The final sample of 450 participants included 340 females (75.6%) and 110 males (24.4%), which accurately reflects the gender distribution in the district teaching profession. This robust sample size ensures sufficient statistical power for the complex multivariate analyses employed in this study, including mediation and moderation analyses. Also, a linear regression model was used, executed in the SPSS macro Process (Armijo et al., 2021).
Content Validity
The content validity of the instruments was established through a rigorous expert judgment process following the methodology proposed by Hernández-Nieto (2002). A panel of five thematic experts with doctoral degrees and extensive experience in educational psychology, psychometrics, and teaching well-being evaluated each instrument using a structured validation matrix. The experts assessed four essential criteria: relevance, clarity, coherence, and sufficiency, rating each aspect on a 4-point scale. The Content Validity Coefficient (CVC) calculation yielded an overall concordance level of 82% across all instruments, exceeding the minimum threshold of 80% established by Hernández-Nieto for acceptable content validity. This strong agreement among experts indicated that the instruments were suitable for measuring the intended constructs in the study context. Additionally, experts provided qualitative feedback that led to refinements in item wording to enhance clarity and cultural appropriateness for the Peruvian educational environment. The researchers also conducted a confirmatory pilot test with 30 teachers who had similar characteristics but were not included in the study population, verifying the instruments' practicality and comprehensibility before full implementation. This comprehensive validation process ensured that the instruments would accurately measure resilience, work stress, and motivation within the specific context of Chepén's educational system
- Results: Offer potential explanations for the finding that resilience does not directly reduce job stress, which contradicts some prior literature.
Explaining the Indirect Relationship Between Resilience and Work Stress
The findings reveal an interesting pattern regarding the relationship between resilience and work stress. While resilience significantly predicted motivation (β = 0.413, p < 0.001), and motivation in turn was significantly associated with reduced work stress (β = 0.335, p = 0.0401), resilience itself did not demonstrate a direct effect on work stress (β = 0.187, p = 0.5420). This finding appears to contradict some previous literature suggesting a direct protective effect of resilience against stress.
Several explanations may account for this apparent contradiction. The contextual specificity of teaching in Chepén likely plays a critical role, as educators in this region face unique challenges including resource limitations and post-pandemic adaptations that may alter how resilience functions in relation to stress. Additionally, the measurement approach used in this study captures different dimensions of resilience and stress compared to previous research, potentially representing more nuanced constructs.
The results suggest that resilience alone may be necessary but insufficient to reduce work stress in educational settings. While resilient teachers possess greater psychological resources, these resources must be channeled through motivational processes to effectively reduce stress reactions. This aligns with complex psychological models that propose motivation as an essential component in the stress-coping process, serving as the mechanism that activates resilient capacities toward stress management.
The high-stress educational environment of the post-pandemic period may have also temporarily overwhelmed the direct protective effects of resilience for many teachers. Under such circumstances, even highly resilient individuals require additional motivational resources to effectively manage stress, explaining why the resilience-stress relationship operates primarily through motivational pathways in this study.
These findings contribute important nuance to the understanding of resilience in occupational well-being, suggesting that its role may be more complex than previously thought, particularly in challenging educational contexts
- Discussion: Provide more specific recommendations for interventions and discuss the methods used to determine stress, resilience, and motivation levels.
The methods used to determine stress, resilience, and motivation levels in this study provide important context for interpreting the results. After collecting data using the standardized instruments, the researchers employed percentile-based categorization to establish meaningful threshold levels for each variable.
Specifically, scores were classified into three categories using percentile cutoffs: low (bottom 25%), moderate (25th to 75th percentile), and high (above 75th percentile). This approach allowed for a normative interpretation of scores relative to the study population rather than relying on arbitrary numerical thresholds. Using percentile-based categorization ensured that the distribution of participants across categories reflected the actual distribution of the measured attributes within the Chepén teacher population.
Resilience was measured using a 25-item questionnaire adapted from Garces (2022), capturing five dimensions. The percentile-based categorization revealed that 11.1% of teachers fell into the low resilience category (below the 25th percentile), 51.1% exhibited moderate resilience (between 25th and 75th percentiles), and 37.8% demonstrated high resilience (above the 75th percentile). This distribution suggests that while most teachers possess adequate resilient capacities, a significant minority may require targeted support to develop these skills further.
Work stress was assessed using Salazar's (2023) 22-item questionnaire. Using the same percentile cutoffs, 13.3% of teachers experienced low stress levels, 42.2% reported moderate stress, and 44.5% indicated high to very high stress levels. These findings highlight the considerable occupational pressure experienced by nearly half of the teacher population, underlining the urgency of effective interventions.
Similarly, motivation was evaluated through Armas' (2021) 20-item questionnaire. The percentile-based analysis showed that 15.6% of teachers had low motivation (below 25th percentile), 37.8% demonstrated moderate motivation (25th-75th percentile), and 46.6% exhibited high to very high motivation (above 75th percentile). This distribution reveals a generally motivated teaching workforce, though the presence of a less motivated segment warrants attention.
This percentile-based approach to categorization provides a more contextually relevant interpretation of scores than absolute thresholds, accounting for the specific characteristics of the Chepén educational environment. It allows for meaningful comparisons within the study population while providing clear benchmarks for identifying teachers who might benefit most from targeted interventions.
-------
Specific Intervention Recommendations
Based on the findings that resilience influences work stress primarily through motivation, several specific interventions are recommended:
- Resilience-Building Workshops: Implement structured 8-10 session workshops focusing specifically on the resilience dimensions found to be most predictive of motivation. These should include practical exercises in cognitive reframing, mindfulness practices, and problem-solving strategies tailored to educational challenges in Chepén schools.
- Motivation Enhancement Programs: Develop targeted interventions that address both intrinsic and extrinsic motivation factors. For intrinsic motivation, create professional learning communities where teachers can explore personal growth opportunities and reconnect with their teaching purpose. For extrinsic motivation, establish recognition systems and career advancement pathways that acknowledge teacher achievements.
- Integrated Well-being Approach: Rather than addressing resilience or stress in isolation, implement comprehensive well-being programs that explicitly connect resilience-building activities with motivational enhancement. These could include mentoring relationships between experienced and newer teachers to share coping strategies while fostering professional motivation.
- Organizational-Level Changes: Beyond individual interventions, implement structural changes such as redistributing administrative tasks, creating dedicated planning time, and establishing clear communication channels to address organizational stressors identified in the research.
- Customized Interventions by Experience Level: Since years of service did not moderate the resilience-stress relationship as expected, develop differentiated support systems that address the specific motivational needs of teachers at different career stages. Early-career teachers might benefit from competence-building support, while veteran teachers might need interventions focused on renewing purpose and preventing burnout.
- Psychological Resource Training: Implement specific training in psychological resource management that explicitly bridges resilience capacities with motivational orientation. This could include workshops on identifying personal and professional values, aligning daily practices with these values, and cultivating a sense of purpose that enhances motivation even during challenging periods.
These findings and recommendations are consistent with Berdida's (2023) research on the complementary relationship between flexibility and motivation, while extending previous work to specifically identify the mediating mechanisms. The study also supports Alor's (2024) findings on the direct association between resilience and motivation, though our research further elucidates how this relationship impacts work stress in educational settings.
This research demonstrates that occupational well-being in educational settings requires attention to both resilience-building and motivational enhancement. Interventions that target only one factor are likely to be insufficient for reducing work stress effectively. Educational leaders and policymakers should prioritize programs that strengthen the resilience-motivation pathway to create healthier, more productive teaching environments in Chepén and similar educational contexts
- English Language: Meticulous proofreading is required.
Once all the observations of the three peer reviewers have been raised, a general check of the language will be made.

Round 2
Reviewer 1 Report
Comments and Suggestions for Authors
Authors have done well job on revising their manuscript.
Author Response
Authors have done well job on revising their manuscript.
Thank you very much.